# Market-Based Governance in Farm Animal Welfare—A Comparative Analysis of Public and Private Policies in Germany and France

**DOI:** 10.3390/ani9050267

**Published:** 2019-05-22

**Authors:** Colette S. Vogeler

**Affiliations:** 1Institute for Political Science, University of Heidelberg, Bergheimer Straße 58, D-69115 Heidelberg, Germany; colette.vogeler@ipw.uni-heidelberg.de; 2Comparative Politics and Public Policy, University of Braunschweig, Bienroder Weg 97, D-38106 Braunschweig, Germany

**Keywords:** agricultural policy, farm animal welfare, market-based governance, governance of common goods, food labelling, animal welfare labels, policy change

## Abstract

**Simple Summary:**

Farm animal welfare policies are in transition: whereas agricultural policy is traditionally characterised by strong state steering, market actors are getting involved in this field. This study aims to improve understanding of these dynamics in the field of farm animal welfare. By conducting a comparative case study of public and private farm animal welfare policies in Germany and France, the findings illustrate how retailers are assuming a leading role in the field. By introducing animal welfare labels and purchasing guidelines, retailers react to rising societal concerns for the welfare of farmed animals. Governmental actors, conversely, are exercising restraint and engage in voluntary rather than regulatory policies. Contrary to the traditionally strong role of the state in agricultural policy, the contribution indicates a shift towards market-based governance in the field of farm animal welfare.

**Abstract:**

The intensification of livestock production and the focus on economic gains of agricultural policy have resulted in animal welfare related challenges. In many countries the societal concern for the welfare of farmed animals is increasing. Whereas policymakers on the European Union’s level and in EU member states have passed specific farm animal protection laws, the existing policies do not always guarantee the welfare of farmed animals. At the same time, the engagement of market actors in the field is increasing. This article explores the development of public and private policies in two countries with very different levels of regulation. By conducting a comparative analysis of public and private policies in Germany and France, the findings illustrate that, although they have different starting points, retailers in both countries are getting increasingly involved in farm animal welfare. In addition, there is evidence that governmental policies are shifting from regulatory to voluntary approaches in cooperation with the private sector. Given that in both countries these dynamics are a very recent development, it remains to be seen whether governmental actors will (re-)assume the lead in the field, whether they will engage in cooperation with private actors, or whether they will leave the task of agricultural restructuring to the market.

## 1. Introduction

Agricultural policy in the European Union is historically characterised by strong governmental intervention. The financial means provided by the Common Agricultural Policy (CAP) have major steering functions. Traditionally, the CAP emphasised food security and ensured that farmers received a steady income [1,2,3]. The focus on economic criteria has fostered high levels of intensification in the farming sector. In livestock production this has resulted in the increasing specialisation and “industrialisation” of modern farming; herd sizes and stocking densities have grown considerably and farmers are under constant pressure to increase productivity. These processes have partly contributed to challenges related to animal welfare [4,5]. Economic incentives for farmers to enhance animal welfare are comparatively low in current animal production processes [6]. Policymakers at European and national levels have reacted by adopting specific farm animal welfare regulations [7,8]. Existing EU animal welfare regulations demand only minimum standards and do not cover all farmed species. Consequently, several member states have passed additional national regulations, which has resulted in a large heterogeneity of animal welfare regulations within the European Union [9,10].

In recent years, the societal concerns for the welfare of farmed animals have increased. Although the level of concern varies between EU member states, surveys reveal that the level of attention paid to the welfare of farmed animals has generally risen [11]. Policymakers seem to address these societal demands for improved farm animal welfare only to a limited degree. The majority of regulatory activities in this field was passed in the 1980s and 1990s [12]. Research also suggests that farm animal welfare is being increasingly addressed by private actors; large retailers in particular are getting involved in this policy field [13]. In addition to business activities that can best be characterised by theories on corporate social responsibility, we now find new patterns of cooperation between private and public actors. Theoretically, the study builds on the literature on co-governance [14,15,16,17]. Co-governance implies the emergence of new forms of cooperation, especially in the governance of common goods, which range from cooperation between private and public actors, to competition and conflict [15].

By reviewing current developments in farm animal welfare policies in Germany and France, this paper explores the interplay of public and private farm animal welfare policies in the two countries. Whereas both countries are major livestock producers and the level of societal concern for the welfare of farmed animals is high, they are most different with regard to the current level of regulation and the development of farm animal welfare policies [9]. Germany has comparatively comprehensive regulations on farm animal welfare, whereas France has only passed regulations as a reaction to binding EU demands [10]. Given these different starting conditions, the study aims to explore whether there are similarities in the current evolution of public and private policies as a reaction to the rising societal attention paid to farm animal welfare.

The findings of the case studies reveal that, despite the different initial conditions in both countries, private actors are increasingly engaging in farm animal welfare whilst governmental actors are acting with restraint. Retailers in particular are setting their own animal welfare standards by introducing animal welfare labels, thereby obligating farmers to provide animal welfare standards that go beyond legal requirements. Whilst the involvement of the retail sector in farm animal welfare has been ongoing in countries such as the UK and Switzerland for over a decade [13,18,19], the contribution on hand shows how these developments are currently intensifying in Germany and in France. In addition to the engagement of private actors, new forms of cooperation that can be systematised by drawing on the literature on co-governance are uncovered in both countries. Most significantly, the market now seems to be assuming a leading role. Moreover, there is evidence that public policies are shifting from regulatory to voluntary approaches in cooperation with the private sector. It is hypothesised that even if agricultural policy has traditionally been the policy field with the greatest state involvement, current developments are changing this logic towards more dynamic forms of co-governance.

The article begins with an overview of the state of the research on co-governance (Section 2.1). The deduced hypotheses will then be tested against two country studies, namely Germany and France. The second section provides a literature review regarding the role of private actors in dealing with animal welfare in livestock farming and links these debates to the literature on co-governance (Section 2.2). The ensuing Section 3 compares the development of public and private farm animal welfare policies in the cases of Germany and France and reveals changing dynamics and a shift towards new and dynamic forms of co-governance. In Section 4, the empirical and theoretical conclusions are discussed and the needs for future research outlined.

## 2. Theoretical Background and Literature Review

### 2.1. Theoretical Background—Co-Governing Common Goods

The involvement of non-state actors in policymaking is at the core of the governance literature. When governments fail to provide or secure common goods, private actors may step in and provide solutions for problems perceived as such by societal actors. The broad body of literature on governance always implies some sort of cooperation between public and private actors in how they address a defined societal problem [20,21]. Within the literature on governance, a specific strand applied to the governance of common goods is the literature on co-governance [14,15,16,17]. By interpreting farm animal welfare as a common good, the study on hand aims to contribute to this emerging debate. It is assumed that this theoretical lens can deepen the understanding of current developments in the field of farm animal welfare policy. While a growing number of empirical studies are already emphasising the growing role of private actors in governing farm animal welfare [13,22,23,24], there is a lack of research that systematically links these findings to the governance literature. The study on hand proposes that, in farm animal welfare policy, we not only find an increase in private standards but also forms of cooperative co-governance.

From a theoretical point of view, co-governance entails a dynamic interaction between public and private actors and distinguishes between different forms of relationships between public and private actors. Contrary to other governance forms, such as corporate social responsibility [25], which is limited to business actors, co-governance always implies a level of interplay between public and business or societal actors [14]. Recent empirical work suggests that co-governance, though mostly cooperative, can likewise be competitive or conflictual, and that the relationships between public and private actors can change over time [15]. For example, several empirical cases have revealed that private standards are sometimes transformed into public standards, and vice versa [26,27,28]. The reasons why private actors get involved in the governance of common goods are manifold: private actors may assume a leading role in the case of government failure or in case policymakers fail to address societal concerns or interests at all. Failure of the state to address such challenges may be accredited to a number of reasons, which can include high transaction costs or the persistent influence of interest groups that want to maintain the status quo [29]. Business actors, conversely, may opt for co-governance if there is a business case for their efforts [15]. Whereas this dynamic interplay of state and market activities has been studied for different policy fields such as forestry or organic agriculture, co-governance in the field of farm animal welfare remains comparatively unexplored. Given the rising societal concerns for the welfare of farmed animals, partly coupled with the willingness to reward higher animal welfare financially [11], we expect that private actors have high incentives to get involved in this policy field. Farm animal welfare therefore presents an enlightening case for studying this emerging interplay of public and private actors. In particular, the study explores how public and private actors engage with each other and aims to draw primary conclusions on their mutual interaction. Do we find cooperation, competition or even conflict in the joint governance of farm animal welfare? In the following section a review of the literature on the engagement of private actors in the field of farm animal welfare policy is presented. By linking this literature to the perspective of co-governance, hypotheses for the ensuing case studies are derived.

### 2.2. Co-Governance in the Field of Farm Animal Welfare Policy

Agricultural policy has historically been characterised by a strong state involvement. In the European Union especially, the Common Agricultural Policy has held major steering functions. Primary policy aims have traditionally been the increase of productivity and the safeguarding of farmers’ incomes. The so-called exceptional policy paradigm assumes that public actors are key in securing these aims [30,31]. This paradigm is recently being challenged by new ideas and interests [3]. This post-exceptional paradigm includes new ideas that emphasise environmental sustainability, climate change, rural development, or ethical considerations and leads to the broadening of the policy agenda [2,32]. Among these new challenges within agricultural policy, farm animal welfare is becoming more important [5,33,34,35]. In many European countries, public concern for farmed animals is rising: 82% of Europeans believe that their welfare should be better protected. Moreover, a majority states that, besides the provision of safe and healthy food, ensuring the welfare of farmed animals should be a top priority for farmers. An average of three quarters would agree to reduce subsidies for farmers in case animal welfare standards are not respected. In both Germany and France, over 90% of respondents would find a reduction in subsidies justified [11]. From a policy analytical perspective, this raises an interesting question, namely: who addresses these apparently high concerns for the welfare of farmed animals? On first consideration, we would expect traditional actors within the agricultural subsystem to pick up the subject of animal welfare. These would be farmers’ associations and political parties. However, research on post-exceptionalism in agricultural policies suggests that confidence in the market’s ability to provide society’s desired outcomes in agricultural policy is increasing [36]. Recent studies point to an increasing role of the retail sector in governing animal welfare, delegating responsibility for animal welfare to the consumer [13,22,23,24]. For other policy fields it was shown how market actors pick up societal demands by adapting their product range to correspond to consumer demands, e.g., by introducing specific standards and labels or by engaging in voluntary agreements [37].

The literature on co-governance assumes that private actors may choose to engage in the governance of a specific policy field for several reasons (see Section 2.1). Given that co-governance is associated with costs for non-state actors, business actors are especially likely to engage in co-governance when they see a business case for their efforts. Secondly, non-state actors may decide to get involved if there is a societal demand that public actors fail to address. Because agricultural policy is traditionally a policy field with a high degree of state intervention, we would not expect private actors to assume sole responsibility, such as in the case of corporate social responsibility. Instead, we would expect private actors to get involved in the field in a complementary manner. Accordingly, it is hypothesised that the rising public concern in many European countries will lead to the emergence of new forms of co-governance in the field of farm animal welfare.

An example of the successful intertwining of state and market activities in the field of farm animal welfare is the labelling of unprocessed eggs in the EU, which denotes the housing system of the hens. Although the inclusion of these labels has been compulsory since 2004, the choice over animal welfare is left to the consumer during purchasing. The four-tiered labelling scheme is related to different housing systems, which consumers associate with different welfare levels of the laying hens [38]. Higher welfare is connected to a price premium. The steering function is thereby appointed to the market and specifically the consumer [23]. On the EU level, there are no comparable initiatives for other animal products, such as meat or milk products. In 2009, the European Commission provided a detailed report on the labelling of further animal products, concluding that the EU would abstain from proposing additional mandatory labels [39].

Some researchers have noted a shift towards the restructuring of the agricultural sector by retailers. Given their market power, retailers are able to govern in different fields, among them animal welfare [40]. This is turning animal welfare into an element of competition between market actors [22]. Generally, different options exist for retailers to get involved in the field of farm animal welfare. Firstly, retailers can introduce specific purchasing guidelines that include criteria related to animal welfare. Examples include the commitment of some retailers, e.g., to the use of cage-free eggs only in processed products or to the prohibition of live plucking for feather products [41]. These criteria can either apply to the whole product range or to selected product lines, which are then often priced at a premium. Secondly, retailers can label selected products according to previously defined animal welfare criteria; e.g., products that come from animals with outdoor-access [23]. Labels can be developed either on the initiative of the retailer or in cooperation with animal welfare organisations, producers or the state [42]. These voluntary labels aim at product differentiation and assign the task of supporting improvements in animal welfare to the consumer. Lundmark et al. argue that animal welfare as a traditionally public good is thereby being increasingly commodified as a private good [13]. Similar observations are made by Maciel and Bock: in their case study of the Netherlands, they find that, as a consequence of new collaborations of retailers and partly of NGOs, “animal welfare is changing from a state-centred to a market-centred policy domain” [24]. This increasing “commodification” is likewise described by Degeling and Johnson [43].

The implications of this shift to the market are controversial. An advantage compared to state regulation is the higher flexibility of private actors to adapt quickly to changing societal demands [44]. Moreover, successful labels may create upward pressure and thereby contribute to a general rise in animal welfare standards; an example of this is the British Freedom Food label [22]. Likewise, animal welfare labelling can pose an advantage for farmers if they are able to achieve a differentiation in the market and a price premium for their products. On the other hand, the flexibility of retail labels may entail uncertainty for farmers regarding which housing systems to choose and invest in. Depending on the scope of the measures required for producing a certain label, smaller farms may be at a disadvantage compared to bigger farms, which have more adequate financial means. The unequal power relation between retailers (in most European countries the retail sector is dominated by a handful of corporations) and farmers may further result in pressure on farmers to participate in the voluntary labels or schemes proposed by retailers.

A second key challenge is the selection of welfare criteria given the diverging interpretations of animal welfare among different actors [4,44]. Usually, consumers have very limited knowledge on animal production and on the respective needs of different farmed species. Nonetheless, private standards focus mostly on the expectations of consumers [13]. In the field of farmed animals, “naturalness” is a common expectation, which is, however, difficult to assess scientifically and objectively [4]. Outdoor-access or pasturing are generally associated with the natural behaviour of animals, which is why consumers are often willing to pay a price premium for products that stem from animals enjoying these. As a consequence, private—as well as state—welfare labels may be based on criteria that meet consumers’ expectations rather than on the animal’s needs. Furthermore, the societal awareness of welfare problems differs with regard to the farming of different species; e.g., the awareness of problems in pig farming is considerably higher than in dairy farming. Therefore, retail initiatives often focus on species whose welfare is deemed in high need of improvement [45].

A third major challenge with respect to the actual improvement of animal welfare by means of welfare labels is the dualisation of the production [23]. Whilst consumer-based instruments may be appropriate for establishing a market for high animal welfare products, in an internationally integrated livestock industry a large share of production is either exported or processed. Obtaining price premiums for these products is challenging [46]. Solely consumer-based instruments in export-oriented countries, such as Germany and France, would then result in divided production: high animal welfare standards in barns that produce for the local market and low standards in barns that produce for the foreign market, for the processing chain, or for the similarly high percentage of consumers unwilling to pay more for animal welfare [44].

A further problem is the lack of uniformity in private animal welfare labels between different retailers. The ever-growing number of labels based on different criteria makes it increasingly difficult to distinguish between them. The efforts of different market players to address societal concerns and make them an element of competition has resulted in a highly fragmented European market for animal-friendly products [22].

Finally, a major challenge to private labels is the trust in these labels as well as the control and enforcement of the underlying standards [42]. In some cases, the credibility of the labels is enhanced by delegating control to independent animal welfare organisations. Another approach to ensure the compliance with standards is to lubricate the cooperation between state and market by producing a state-initiated but voluntary animal welfare label. This option is currently being discussed in Germany for meat products [47].

Building on this review of the state of the research on the engagement of private actors in animal welfare, the following section will explore the development of private sector initiatives and their interrelation with state activities in the countries of Germany and France. Richards et al. argue that retailer-driven agricultural restructuring generally varies depending on a country’s “neoliberal” orientation [40]. Lever and Evans consider farm animal welfare as a central feature of corporate social responsibility activities, which are more pronounced in neoliberal markets [48]. Following these assumptions, Germany and France, which both share a coordinated market economy with a traditionally strong state involvement, present at first sight similar cases, since the market plays a minor role in them [45]. On the other hand, the role of the state in the field of farm animal welfare varies considerably, for the legislation in Germany is advanced compared to the underdeveloped national regulation in France [9,49]. The comparison is guided by the literature on co-governance. It will first examine whether the engagement of private actors is increasing; and secondly, how the relationships between public and private actors evolve: are they cooperative, competitive or even conflictual? Building on the research of the increasing commodification of farm animal welfare, the study aims to explore whether the engagement of private actors can be shown for countries as different as Germany and France: are there similarities in the development of farm animal welfare policies and in the relationship between public and private actors, despite the different starting conditions? To this aim, the case study section begins by comparing regulatory policies and current policy changes, followed by an analysis of private policies. In the discussion, the development of public and private policies and the relationships between these are investigated.

## 3. Case Studies

### 3.1. A Comparison of Farm Animal Welfare Regulations in Germany and in France and the Impact of EU Membership

Germany and France are among the biggest producers of animal products in the European Union and in both countries the livestock sector significantly contributes to agricultural output [50,51]. As members of the European Union, Germany and France must comply with the EU directives on farm animal welfare. Although there is no uniform definition of animal welfare, farm animal welfare on the European Union’s level is inspired by the so-called “five freedoms” [12]. The five freedoms have been developed following a report on livestock husbandry that was commissioned by the British government in 1965. The so-called Brambell report uncovered wide-spread suffering in modern industrial farming. As a consequence the Farm Animal Welfare Council proposed that farmed animals should at least be granted the following freedoms: (1) freedom from thirst, hunger or malnutrition; (2) appropriate comfort and shelter; (3) prevention, or rapid diagnosis and treatment, of injury and disease; (4) freedom to display most normal patterns of behaviour, and (5) freedom of fear [52]. The five freedoms have since then been adopted by governmental and non-governmental actors to assess the welfare of farmed animals and to develop policies and husbandry systems that ensure these freedoms. In 2009, the British Farm Animal Welfare Council (FAWC) proposed an expansion of the five freedoms in response to the criticism that the original five freedoms concentrate solely on suffering and needs. The FAWC defined the policy goal that all farm animals should have a life worth living and an increasing number should have a good life [53]. Despite these ambitious goals, the ensuing analysis will show that current regulations at the EU level and in the two countries under analysis hardly manage to comply with the original five freedoms, let alone with the addition from 2009.

In the 1970s, the EU passed regulations to protect animals in slaughterhouses (1974) and during transport (1977). Then in the 1980s, specific regulations for rearing pigs, calves and laying hens were introduced, followed by ones in 2007 for chickens kept for meat production [54]. These directives set minimum standards for the rearing and handling of farmed animals, e.g., by setting minimum cage sizes for the individual species. For other species, only non-binding recommendations exist. Member states are responsible for implementation and control. In 1997, the recognition of animals as sentient beings was included in the Treaty of Amsterdam [55]. A common problem in EU animal welfare policies is the implementation and enforcement in member states. Failure to enforce EU regulations has resulted in hundreds of proceedings and over twenty court cases to date [56]. In France especially, the enforcement of EU regulations has been weak [54]. The insufficient implementation and the late prohibition of confined housing systems, such as battery cages for chickens, veal crates and dry sow stalls have resulted in several EU proceedings against France [57,58]. Although singular EU proceedings have likewise been made against Germany, the level of national regulations is, in this case, only partly a consequence of EU membership (see the overview in Table 1) [9,49].

In addition to the regulatory policies, the EU provides financial assistance based on support schemes from the Common Agricultural Policy (CAP) for farmers who take animal-welfare-related measures. Nonetheless, out of the total CAP budget, only 1.4% is spent on such measures. Financial incentives are provided for standards in animal production that go beyond the mandatory standards defined by EU regulations. The EU directly funds animal welfare through the rural development measure 14, “animal welfare payments”, in the second pillar of the CAP. In the reporting period 2014–2020, there were large differences between the planned farm animal welfare spending among member states. Interestingly, France did not plan any animal welfare spending within this measure, whereas Germany is scheduled to spend almost 100 million Euros, the sixth largest amount among the member states [54]. An additional EU farm animal welfare policy is the compulsory labelling of unprocessed eggs to denote the housing system. Since 2004, unprocessed eggs must carry a label that indicates whether the laying hens live in cages or in other rearing systems [38]. This policy represents an intertwining of state and market activities and appoints the responsibility for animal welfare to the consumer. The share of eggs stemming from cage systems has since then declined, though with significant differences between member states. In Germany, only 10% of laying hens are kept in cage systems, whereas in France still 69% are kept in enriched cages [59] (conventional cages have been phased out by EU law already). Building on this summary of the impact of EU membership on national animal welfare policies, a brief introduction into the peculiarities of German and French farm animal welfare policies is presented. In the subsequent section, recent policy changes in the two countries are analysed.

#### 3.1.1. Farm Animal Welfare Policies in Germany

Unlike other European member states, Germany often goes beyond the EU directives for animal welfare regulations, which in many cases are elaborated in great detail [9]. Although there are exceptions with regard to different species, legal standards in Germany are among the highest in the world; more detailed regulations only exist in a few countries, such as the United Kingdom, New Zealand and Switzerland [10,18]. The first animal welfare regulations were passed in Germany in 1933. In 1972, the animal protection law was adopted and in 2002 animal protection was even included in the German constitution as a national objective [60]. Today, detailed regulations for the different farmed species exist (see Table 1). For poultry, laying hens, pigs, calves, dairy cows, rabbits and fur animals, species-specific regulations regarding their keeping have been passed. On the contrary, regulation is lacking for other livestock species, such as sheep, goats, beef cattle, ducks and geese. In some cases, policymakers have opted for voluntary agreements instead of regulations; e.g., a voluntary agreement between the government and the poultry farmers’ association sets common standards for the housing systems and rearing of turkeys [18]. Despite the comparatively advanced level of regulations, it is highly controversial if the existing laws sufficiently protect farmed animals [46,61].

Within the framework of the German animal protection law, practices are permitted even though they fail to meet the five freedoms. An example currently the object of much debate is the use of farrowing crates for sows (which is common in most countries with intensive pig production). These crates restrict the movement of the sow to the degree that it cannot turn around at all or lie down properly. A second welfare challenge in the field of pig farming is the castration of male pigs without anaesthesia, which was prohibited by law in 2013 with a transitional period until 2019 [62]. Given the opposition of farmers against the regulation, the German parliament in 2018 extended the transitional period for castration without anaesthesia for two more years on the grounds that there were no economically feasible alternatives for the castration process. The case illustrates the subordination of animal welfare under economic interests. The German animal protection law states that no person may harm an animal without “good or reasonable cause”. As a consequence of this formulation, economic interests are often interpreted as a reasonable cause before court. The superiority of economic variables is also manifest in the common procedure of slaughtering male day-old chicks in egg production. This practice was taken to court in 2016. However, the court ruled that the economic interests of farmers were superordinate to animal welfare interests [63]. In addition to these challenges, problems with implementation and enforcement of animal welfare laws have been documented and have repeatedly been the subject of news coverage. The enforcement of animal welfare regulations lies in the responsibility of the federal states. Controls are least likely in Bavaria, where livestock farmers are controlled on average every 48 years, whilst in Lower Saxony, the state with the highest regional density of livestock units, farms are controlled on average every 21 years [64].

#### 3.1.2. Farm Animal Welfare Policies in France

In France, regulatory animal welfare policies essentially correspond to the minimum requirements put forward by the European Union [10]. Legislation has only been passed to transfer EU directives into national law, e.g., concerning the prohibition of battery cages for chickens and of veal crates (see Table 1). In addition to the lack of additional animal welfare legislation, EU regulations have often been transferred with delay to the national context, and the EU has repeatedly initiated proceedings against France for non-compliance with EU demands [57,58]. Influenced by particularly powerful farmers’ associations, animal welfare in France is mainly considered a constraint for agricultural productivity. Interestingly, there is a high concern and growing public interest in the welfare of farmed animals among French citizens [11]. This polarisation regarding farm animal welfare was recently expressed in the political negotiations surrounding the national agricultural policy, the “*loi agriculture et alimentation*” that was passed in 2018 [65]. Although the law contains several aspects that aim at welfare improvements, the proposed policies are limited to harsher penalties for the mistreatment of animals, particularly during transport and slaughter, and to instruments used for surveillance at slaughterhouses. The law does not contain any requirements on housing systems, apart from the prohibition of building new cage-systems for hens, which is, however, the transposition of EU demands and therefore not a national advance.

### 3.2. Policy Change in Germany and France in Farm Animal Welfare Policies

The brief overview of farm animal welfare regulations in Germany and France reveals the significant differences in regulations and in the development of farm animal welfare policies between the two countries. Interestingly, there is a comparable and high level of public concern for the welfare of farmed animals in both cases. Variation exists with regard to the question of who should assume responsibility for the issue. Whereas in Germany over 50% of respondents find that the state is responsible for the welfare of farmed animals, in France only 35% do so. Here, there is greater confidence that both public and private actors should care for the issue, with over 50% of respondents believing so [11]. In the following sections, reactions of public and private actors to these societal demands are analysed. Firstly, policy changes within the last two electoral periods in both countries are collected to explore if and how policymakers in both countries react to the depicted societal concerns. The analysis is limited to policies on the national level to enable comparability across cases. For example, the federal states in Germany have the scope of action to pass animal welfare policies by means of decrees, voluntary agreements or financial support [49].

For Germany, four relevant regulatory policies were found within the last two legislative periods; two of these refer to fur farming, one to the slaying of day-old male chicks, and one concerns the slaughtering of pregnant animals. An overview is presented in Table 2. In addition to these regulatory instruments, a voluntary policy with the German poultry association was agreed upon in 2015 regarding the renunciation of beak clipping for laying hens. Secondly, since 2017, the government has made to include a voluntary animal welfare label on meat products. The label has been developed jointly with private actors and presents an example of a cooperative co-governance approach. A government-led scheme would increase the legitimacy and the trust in the label. At the same time, given the voluntary character of the label, the cooperation of the processing chain and of retailers is essential. Initial plans regarding this label were presented in January 2017 by the then-minister of agriculture Christian Schmidt. The minister planned a voluntary label with two levels, namely an entry and a premium level on meat products, and its introduction into the market in 2018. However, the minister did not realise his plans within the remaining governmental period, which ended in autumn 2017. The initiative of the animal welfare label was adopted by the following government and formalised in the coalition contract. In March 2018, Schmidt’s successor in office, Julia Klöckner, announced the introduction of a voluntary animal welfare label. In May 2018, minister Klöckner presented plans for a three-level label based on existing private animal welfare initiatives and specifically on the “*Initiative Tierwohl*”, a cooperation between retailers and producers (see Section 3.3). By March 2019, the proposal was still in the policy process. The legislative process is assumed to be finished by the end of 2019. According to the ministry of agriculture, meat products with the label will be available in supermarkets in 2020 [66].

For France, the only relevant policy changes with regulatory character were introduced in 2018, in the frame of the national law on agriculture and food. The exact measures are compiled in Table 3. Although animal welfare is addressed in several points of the law, the focus of the policy package is on enforcement and control, e.g., the harshening of penalties for offenses against existing laws. The proposed measures do not include regulatory policies beyond what is necessary to comply with European standards or other binding policy instruments [72]. The only regulation concerning housing systems is the prohibition on installing new cage systems for laying hens. This, however, is a fulfilment of an EU directive. In addition to these regulatory policies, a national animal welfare strategy was passed in France in 2016. The strategy addresses current challenges in livestock production, such as castration without anaesthesia and the tail-docking of pigs, the painful practice of beak-trimming in poultry production as well as deficiencies in transportation and slaughter practices. The problems addressed in the national animal welfare strategy do not have regulatory character but build on voluntary approaches, innovations and support schemes. In 2018, the ministry of agriculture presented a plan with priorities for action in the field of farm animal welfare. The strategy again is not a regulatory policy but a publication of the intents and aims of the government.

To summarise the analysis of recent policy changes, it can be concluded that in both countries public actors have been taking more action in the field of farm animal welfare in the last legislative periods. Despite the different starting points (regulations in place), there seems to be a focus on voluntary policies in both countries. Policymakers are cautiously reacting to the equally high societal concerns in both countries by identifying needs for action, by elaborating animal welfare strategies (as in the case of France), or by introducing voluntary but state-steered labels (as in the case of Germany). Most importantly, the identified measures have voluntary or cooperative character. 

### 3.3. Private Actors’ Engagement in the Field of Farm Animal Welfare

In the following section, the role of private actors will be analysed. Given the fact that policymakers are reacting rather cautiously to the changing societal demands, private actors are likely to step into the field to fulfil particular consumer demands, e.g., if they see a positive business case. To explore if that is the case, the next section compares private animal welfare labels and strategies in the two countries. To delimit the frame of the investigation, the analysis will focus on the initiatives of the retail sector. This approach follows the existing research findings on the growing importance of retailers in farm animal welfare in other countries (see Section 2). To this aim, the private animal welfare labels currently existing in Germany and France have been collected. Interestingly, the involvement of the retail sector in animal welfare has recently reached new heights: Since the beginning of 2018 in Germany, several big retailers, foremost discounters, have introduced animal welfare labels on their meat products [76]. In 2019, the major retailers announced the introduction of a joint animal welfare label to create uniformity and increase transparency for the costumer. Likewise in France, leading retailers announced in 2018 the introduction of a four-stage animal welfare label [77]. An overview is compiled in Table 4. The ensuing section discusses the findings in detail for the two countries.

Since 2015, several retailers in Germany have introduced labels for milk or eggs. Examples are the discounters Aldi Nord and Aldi Süd, which label fresh milk that stems from dairy cows with outdoor-access. With regard to the discussion in Section 2, the label is based on the expectations of consumers regarding naturalness, since outdoor-access is commonly perceived as better animal welfare. In 2016, the retailer Rewe introduced a labelling system for unprocessed eggs that stem from laying hens whose brothers were raised as broiler, instead of being slaughtered as day-old chicks. The retailer therewith addressed an animal-welfare-related concern that was the subject of intensive media and political discussion and steps into a field in which the national government has failed to take regulatory measures.

In addition to the initiatives of individual retailers, the label “*Für mehr Tierschutz*” was introduced in 2013 by a joint cooperation of animal welfare organisations, farmers’ organisations, scientists and retailers. The label offers an entry level that is marked with one star and a premium level marked with two stars. Species-specific criteria have been developed together with scientists for broiler, pigs, laying hens and dairy cows. The criteria refer to the different steps of livestock farming and include guidelines on transportation and slaughter. The nature of these depend on which species the criteria refer to, stocking densities, the availability of manipulatable materials, or the prohibition of non-curative measures, such as dehorning. Labelled products can be found in different retailers across Germany. In addition, some retailers, such as Aldi Süd and Nord, have based their own labels for milk on the criteria developed by this initiative. Complementary to the labelling of products for improved animal welfare, most retailers in Germany have published specific animal welfare purchasing guidelines (for an overview, see the compilation elaborated by the German Albert Schweitzer foundation) [83].

In 2018, new impetus came into the animal welfare activities of German retailers. At the beginning of the year, the retailer Lidl introduced a four-stage labelling scheme for meat products in its stores. The four grades were inspired by the European egg labelling scheme and focussed on keeping conditions. Products labelled with grade 1 stem from animals that live under conditions that fulfil the legal requirements. Grade 2 labels come from animals that enjoy more space and manipulatable material. Grade 3 includes access to outdoor climate areas, GMO-free feeding and more space; and grade 4 denotes adherence to existing organic standards. Following this advance from Lidl, several other retailers have developed labels for meat products. Consequently, there is a multitude of different labels in the market, which in turn has reduced transparency for consumers. Therefore, in 2019 the biggest retailers announced the introduction of a joint four-stage label called “*Haltungsform*” with shared criteria to avoid fragmentation. This new label is combined with and based on the already existing “*Initiative Tierwohl*”. The “*Initiative Tierwohl*” was introduced in 2015. Participating retailers pay a certain amount of money per year that is centrally collected and then re-distributed to the farmers who supply the retailers. Farmers eligible to receive money from the initiative have to fulfil certain criteria regarding animal health and keeping facilities. Species-specific criteria exist for pigs, piglets, sows and poultry; for each species there are compulsory and additional elective criteria. However, products of this initiative are not labelled. The purpose of the initiative is to compensate extra performances by farmers that surpass the legislative standards. Currently, over 4000 pig farmers and almost 2000 poultry farmers (including broiler and turkey) are participating in the “*Initiative Tierwohl*”. In 2018, the initiative disposed over 130 million Euros that were distributed among the farmers.

Interestingly, the recently introduced private label “*Haltungsform*” runs in parallel with the planned government label and is therefore in competition with the state’s attempts. The unilateral approach of the retailers has, until now, been an exclusively private policy, whereas the state label would rely on cooperation.

Similar to the German case, we can observe an increasing involvement of retailers in France in farm animal welfare. Until recently, there was only one animal welfare label worth mentioning in France, the “*Label Rouge*” for chicken products, which was introduced in 1965. The “*Label Rouge*” presents an example of cooperative co-governance and, similar to the planned label in Germany, has voluntary character. The label is granted by public agencies and includes actors from all production steps and from hatcheries to abattoirs. In addition, retailers participate in the “*Label Rouge*” initiative. Animal welfare criteria relate to outdoor-access, slow-growing breeds and specific feeding. The market share of “*Label Rouge*” chicken has reached up to 30% in domestic consumption, but only around 15% in total production due to high export shares [82]. The Label Rouge has recently been expanded and adopted for other species, among them cows, veal and pigs. Welfare criteria relate to outdoor-access or straw bedding and, similar to the poultry sector, to the selection of specific or slow-growing breeds. Contrary to the poultry sector, the market share of pigs raised under the “*Label Rouge*” requirements is far below 5%. The “*Label Rouge*” represents the sole successful example of a public–private cooperation in France in the field of farm animal welfare. In addition to that, there have only been a few solely private sector initiatives. Compared to retailers in the UK, Switzerland or Germany, existing studies reveal that French companies are lagging behind in the development and implementation of animal welfare standards [41]. Nonetheless, recent market activities suggest that farm animal welfare is quickly gaining attention: in 2018, a group of leading retailers announced a specific four-stage animal welfare labelling scheme for chicken products [84]. Interestingly, the label was developed together with animal welfare organisations, a development that suggests the formation of new coalitions in the field. The new label will be divided into four grades based on different welfare criteria that are assumed to cover the entire production process. As with the recently introduced retailer label in Germany, this label only relies on private actors and does not include public actors.

## 4. Discussion and Conclusions

This study aimed to explore the development of public and private policies in the field of farm animal welfare in two countries with very different initial conditions. Whereas Germany has comprehensive animal welfare regulations in place for some species, specific national regulations that go beyond EU demands are absent in France. Nonetheless, in both countries, citizens believe that the welfare of farmed animals should be better protected. The comparison illustrates that, despite their different starting points, market-based governance is increasing in both countries. The very recent introduction of animal welfare labels by leading retailers in Germany as well as in France indicates that animal welfare is becoming a subject of competition in both countries. This is, however, truer in Germany than in France. Retailers are responding therewith to the societal concern for the welfare of farmed animals and to changing consumer demands.

For the case of France, we find growing public attention paid to farm animal welfare, even though this is not reflected in public policies yet. Recently, farm animal welfare issues have moved onto the political agenda—as in the case of the national agricultural law, though they were subordinated to traditional agricultural interests during the policy process. These developments point towards an increasing conflict over the issue as animal welfare is still often subordinated to economic gains in public policymaking. This finding also applies to the German case: in several cases, such as in that of the recent prolongation of the castration of male pigs without anaesthesia, economic interests have dominated over animal welfare. Given the conflict between increasing productivity as a primary goal of exceptional agricultural policies on the one hand and animal welfare concerns on the other, private policies may present an opportunity for the improvement of animal welfare in the production process. For the case of Germany, this dominance of market actors is a rather new development. Even recent public policies have built on voluntary agreements or assistance schemes, instead of on binding regulations, thereby presenting new forms of co-governance. Although the German ministry of agriculture already announced plans for the development of a state farm animal welfare label several years ago, implementation is lagging behind. This state animal welfare label has voluntary character, follows existing market initiatives, and is being introduced at a time when the majority of retailers have already passed their own animal welfare labels. The state thereby seems to resign from its steering function in animal welfare and leaves this task to the market and to the decision of consumers. 

The takeover of farm animal welfare by private actors seems to be taking place in Germany and in France alike—independent of the different initial conditions. The study illustrates that farm animal welfare policies in both countries are in transition and private actors are increasingly getting involved in this field. Contrary to the societal expectations in both countries, governmental actors are acting with restraint. Regulatory policies are scarce, whereas current policies often have voluntary character. Given that the major role of retailers in farm animal welfare is a rather new development—the majority of specific animal welfare labels was only passed in 2018 or 2019 in both countries—it remains to be seen whether the state will (re-)assume the lead in the field or, on the contrary, leave the task of agricultural restructuring to the market.

A key finding is the prevalence of private actors’ engagement as compared to cooperative governance between private and public actors. In Germany, the government, by introducing a public, but voluntary, state label, is aiming to establish new forms of cooperative co-governance. The label can be interpreted both as a reaction to public concerns and as a response to the pressure coming from the retail sector that is currently experimenting with private animal welfare labels.

To what extent the recently launched private and public–private animal welfare labels will actually contribute to changes in purchasing behaviour, and consequently to changes in production processes, currently remains unclear. The criticism on the role of retailers in setting their own farm animal welfare standards has been discussed in Section 2. In particular, selecting welfare criteria that satisfy consumers’ expectations is a challenge, when these criteria aim to improve the way farmed animals are treated. A second major challenge relates to the intrinsic unpredictability of the agricultural industry, which means that farmers, who must meet the demands of retailers, are unable to plan far in advance. Livestock facilities and stables are usually run for several decades, and short-term and unpredictable changes represent major financial challenges for farmers. In comparison with public policies, private policies are less reliable, and retailer demands and purchasing guidelines may change relatively quickly. The attempt of the German government to introduce a label in cooperation with public and private actors may strike a balance between public expectations and farmers’ needs. The third major challenge regarding private animal welfare policies refers to the control and enforcement of private standards. In some cases, such as in that of the recently introduced French animal welfare label by the Casino Group, retailers cooperate with animal welfare organisations. In the field of farm animal welfare, successful examples for such cooperation can be found in Switzerland, where the Swiss animal welfare organisation oversees the control of selected private standards.

Future studies should build on these findings and include explanatory variables for the shift towards market-based governance to the analysis. Moreover, international comparison across the European Union could contribute to the identification of generalisable patterns and thereby deepen the understanding of forms of governance in farm animal welfare policymaking.

## Figures and Tables

**Table 1 animals-09-00267-t001:** Regulatory farm animal welfare policies in Germany and France.

Regulations	EU	Germany	France
Slaughter	x	x	x
Transportation	x	x	x
Laying Hens	x	x	x
Poultry	x	x	x
Sows	x	x	x
Pigs	x	x	x
Beef Cattle			
Dairy Cows			
Calves	x	x	x
Sheep, goats			
Rabbits		x	
Ducks, geese			
Fur animals		x	

Source: own compilation.

**Table 2 animals-09-00267-t002:** Policy changes in farm animal welfare in Germany.

Date	Title of the Bill	Content
2017	Entwurf eines Gesetzes zur Änderung futtermittelrechtlicher und tierschutzrechtlicher Vorschriften [67]	1. Tightening of the requirements for keeping fur animals, with the aim of making fur farming economically unfeasible.
		2. Prohibition of the slaughtering of animals in the last third of pregnancy.
2015	Entwurf eines … Gesetzes zur Änderung des Tierschutzgesetzes [68]	Change of the animal protection law to forbid the slaying of day-old chicks, which is commonly undertaken to increase the economic feasibility of egg production. The bill is a reaction to a federal’s court rule.
2015	Entwurf eines … Gesetzes zur Änderung des Tierschutzgesetzes [69]	Bill to prohibit fur farming in the long-term (initiative of the Bundesrat).
**Date**	**Voluntary policy**	**Content**
2017	Proposals for a voluntary state animal welfare label [70]	Currently in the coordination process with the federal states and the associations (2018-06-18). First proposed in 2017.
2015	Voluntary agreement with the poultry association [71]	From August 2016, no de-beaking in hatcheries for laying hens.

Source: own compilation.

**Table 3 animals-09-00267-t003:** Policy changes in farm animal welfare in France.

Date	Title of the Bill	Content
2018	Loi Agriculture et alimentation [65]	Extension of the offense of animal abuse during transport and slaughter.
		Increase penalties for animal abuse.
		Possibility for animal protection organisations to take civil action.
		Appointment of animal protection officer in each slaughterhouse.
		Experimentation of video surveillance in slaughterhouses.
		Prohibition on installing new cages for laying hens (EU demand).
**Date**	**Voluntary policy**	**Content**
2018	Plan d’action prioritaire en faveur du “bien-être animal” [73]	Strategies to improve animal welfare during production, transport and slaughter (voluntary measures or improved enforcement); financial support for farmers; agroecological transition.
2016	Stratégie de la France pour le bien-être des animaux [74,75]	

Source: own compilation.

**Table 4 animals-09-00267-t004:** Private animal welfare labels in Germany and in France.

Retailers	Label	Cooperation Partners	Start	Farmed Species	Criteria
**GERMANY**					
Aldi Nord, Aldi Süd, Edeka, Kaufland, Lidl, Netto, Penny, Rewe	Haltungs-form [76,78]	Retailers, Initiative Tierwohl	2019	Broiler, pigs, beef cattle, calves, turkey	Keeping conditions analogous to the EU egg-labelling scheme, Grade 1: Barn: level of existing legislation, Grade 2: Barn plus: more space and manipulatable material, Grade 3: Outdoor climate: more space, GMO-free feeding, access to outdoor climate areas, Grade 4: Organic: level of existing organic legislation.
Aldi Nord, Aldi Süd, E-Center, Edeka, Kaufland, Lidl, Netto, Norma, Penny real, Rewe	Für mehr Tierschutz [79]	Scientists, retailers, animal welfare organisations, farmer’s associations	2013	Broiler, laying hens, pigs, dairy cows	Two grades, entry (1 star) and premium (2 star) grade. Species-specific criteria referring to stocking-densities, outdoor-access, manipulatable material, non-curative measures, transportation times, slaughter guidelines.
**FRANCE**					
Casino Group	L’étiquette bien-être animal [80]	Retailers, animal welfare organisations	2018	Broiler	Four-stage label with a focus on keeping conditions from grade A (standard) to grade D (superior). Species-specific criteria to be defined, e.g., for the areas of housing, feeding, and the provision of manipulatable material. Animal welfare criteria defined for the whole production process
Retailers, producers, state	Label Rouge [81,82]	Public agency, producers, retailer	1965	Laying hens, broiler, pigs, beef cattle, calves	Outdoor-access, specific feeding, partly slow-growing breeds, other non-animal welfare related criteria

Source: own compilation.

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
