# Peer review of "Market-Based Governance in Farm Animal Welfare—A Comparative Analysis of Public and Private Policies in Germany and France"

_animals, 2019, doi:10.3390/ani9050267_

Round 1

Reviewer 1 Report

The paper addresses a very timely and important issue, and represents a niche in the literature on animal concerns by empirically analyzing and comparing public and private policies on animal welfare in two European countries. I have the following major and minor comments that are meant to further strengthen the paper.

Major comments

-       The paper is not positioned in a particular theoretical perspective, nor does it apply a conceptual framework for analysis. With this, the paper remains rather descriptive and falls short of an academic analysis. The author refers to relevant debates on neoliberalism and commodification, governance, and policy fragmentation, but does not apply any of the frameworks developed in the (global environmental) governance literature that these debates are part of. There are e.g. several frameworks to study regime or institutional interaction or interplay, integrative governance, fragmentation, or institutional effectiveness, which would fit really nicely. The paper also does not position the empirical analysis of the paper in the academic debates, as introduced in section 2. I would expect the discussion and conclusion section to do so.

-       The use of two hypotheses seems out of place. They are introduced rather late in the paper, and the conclusion does not discuss whether the hypotheses have been proven or disproven. I would advise the author to delete the hypotheses.

-       The organization of the paper can be significantly improved in order to enhance the line of argumentation and reduce redundancy between the different sections. The description of the two country case studies is currently chopped up into several sections. I would advise the author to bring together the evidence for each country into one section.

-       Section 2 is called Materials and methods, but this is not what this section is about. It is actually a (really interesting) literature review on animal welfare policy and the role of private governance on this issue. A methodology section is lacking. I would advise the author to develop a separate section that explains the applied research methods.

Minor comments

-       line 24. The abstract says the private policy has been developed “as a consequence” of lacking governmental policy, but the paper does not analyze or prove this causal relationship.

-       Line 61 (and elsewhere). I have never seen the term “co-governance” used before. I have seen the terminology of multi-actor governance or collaborative governance, but never co-governance. I think the more general term of “governance” would be better here.

-       Line 79. The term “exceptional” agricultural policy needs to be defined for readers who are not agriculture experts.

-       Line 293. What year was the Eurobarometer published?

-       Lines 290-300. This section can be moved up and integrated into an earlier introduction on shifting public opinion.

-       Line 307 and beyond. Please define “recent”. This can best be done in the methodology section perhaps. As stated in the above, this differentiation between recent and earlier policies seems rather arbitrary, and I would advise the author to integrate the analysis per country.

-       Line 325. “at the time of writing”. Is this still accurate and up-to-date?

-       Line 368. “will focus on initiatives of the retail sector”. I would have expected this much earlier in the paper, either in the introduction or in the methods section.

-       Line 454 and beyond. The discussion on challenges is interesting, but is perhaps better placed in the discussion section.

-       While the paper overall is well written, it could benefit from editing by a native English speaker. 

Author Response

Letter to Reviewer 1

Dear Reviewer,

Thank you very much for your detailed and very constructive review. Your suggestions have made a valuable contribution to the improvement of my study. In addition to the revisions in my manuscript, please allow me to respond to your comments. 

Kind Regards

Reviewer Comment

Author Response

The paper is not positioned in a particular theoretical perspective, nor does it apply a conceptual framework for analysis. With this, the paper remains rather descriptive and falls short of an academic analysis. The author refers to relevant debates on neoliberalism and commodification, governance, and policy fragmentation, but does not apply any of the frameworks developed in the (global environmental) governance literature that these debates are part of. There are e.g. several frameworks to study regime or institutional interaction or interplay, integrative governance, fragmentation, or institutional effectiveness, which would fit really nicely. 

I agree that the theoretical framework has been underdeveloped in the original version of the manuscript. My point of reference is the literature on co-governance, which I believe presents an interesting starting point for the exploration of current developments in the field of farm animal welfare policies. 

To address this concern I have added section 2.1. which presents an overview of the literature on co-governance. This is a relatively recent debate within the governance debate. Co-governance includes the dynamic interplay in the governing of common goods between public and private actors. Contrary to other governance forms such as corporate social responsibility, that is limited to business actors, co-governance always implies the interplay between public and business or societal actors. Recent empirical work suggests that whereas co-governance is mostly cooperative it can likewise be competitive or conflictual and the relationships can change over time. In addition to the theoretical discussion in section 2.1. I have systematically integrated the discussion of co-governance to the empirical analysis. In the application to the empirical cases (farm animal welfare policies in France an Germany) the analysis uncovers an increase in cooperative co-governance in addition to private governance.

The paper also does not position the empirical analysis of the paper in the academic debates, as introduced in section 2. I would expect the discussion and conclusion section to do so. 

The discussion and conclusion have been revised and now include references to sections 2.1. and 2.2. . This positions the findings in the debates on public and private farm animal welfare policies and on co-governance. 

The use of two hypotheses seems out of place. They are introduced rather late in the paper, and the conclusion does not discuss whether the hypotheses have been proven or disproven. I would advise the author to delete the hypotheses.

The reviewer is right that the late use of hypotheses is misleading. In addition to the revision of the theoretical framework, the hypotheses have now been integrated in sections 2.1/2.2. and refer to the emergence of new governance forms in the field of farm animal welfare policies, which is at the core of the contribution. 

The organization of the paper can be significantly improved in order to enhance the line of argumentation and reduce redundancy between the different sections. The description of the two country case studies is currently chopped up into several sections. I would advise the author to bring together the evidence for each country into one section. 

To improve the structure of the manuscript I have added subsections to the analysis which I believe also improves the reading flow. However, I have decided to remain with the separation of the case study in two major parts: one section that compares public policies in the two countries, and one section that compares private policies in the two countries. In my opinion this structure underlines the comparative design of the manuscript. 

 Section 2 is called Materials and methods, but this is not what this section is about. It is actually a (really interesting) literature review on animal welfare policy and the role of private governance on this issue. A methodology section is lacking. I would advise the author to develop a separate section that explains the applied research methods. 

The reviewer is right that the title of section 2 was misleading. I have now renamed section 2 (Theoretical Background and Literature Review) and in addition I have revised the section that is now separated in two subsections: 2.1. Theoretical Background – Co-Governing Common Goodsand 2.2. Co-Governance in the Field of Farm Animal Welfare Policy.

I have explained the materials and methods used in section 3 (the section that includes the empirical analysis) because I believe that this will improve the reading flow and help the reader to follow the empirical analysis. 

Minor comments

line 24. The abstract says the private policy has been developed “as a consequence” of lacking governmental policy, but the paper does not analyze or prove this causal relationship.

The reviewer is right that this formulation is misleading, I have therefore adapted it to clarify the aim of the analysis, which is the exploration of new governance forms in the field of farm animal welfare policy in Germany and France. 

Line 61 (and elsewhere). I have never seen the term “co-governance” used before. I have seen the terminology of multi-actor governance or collaborative governance, but never co-governance. I think the more general term of “governance” would be better here. 

I agree that the term co-governance has not been explained sufficiently in the original version of the manuscript. I have addressed this concern in section 2.1. .

Line 79. The term “exceptional” agricultural policy needs to be defined for readers who are not agriculture experts.

In section 2.2. a short explanation of exceptionalism and post-exceptionalism has been added. Exceptional agricultural policies focus on the increase of productivity and the safeguarding of farmer’s incomes. Within this paradigm it is assumed that governmental actors are key in securing these aims. This paradigm is recently being challenged by new ideas and interests: So-called post-exceptionalism includes new ideas emphasizing environmental sustainability, climate change, rural development or ethical considerations and as well issues concerning farm animal welfare

Line 293. What year was the Eurobarometer published? 

The year of publication (2016) has been added to the text and to the literature and illustrates the high topicality.

Lines 290-300. This section can be moved up and integrated into an earlier introduction on shifting public opinion. 

I have now also referred to these data earlier in the manuscript. However, I think it is important to mention the differences in public opinion in this section as well to guide the reader through the analysis and to understand why business actors have incentives to get involved in the field of farm animal welfare.

Line 307 and beyond. Please define “recent”. This can best be done in the methodology section perhaps. As stated in the above, this differentiation between recent and earlier policies seems rather arbitrary, and I would advise the author to integrate the analysis per country. 

I agree that the formulation “recent” is a bit arbitrary. The analysis actually focuses on the last two legislative periods in the two countries. I have explained this in the manuscript now.

Line 325. “at the time of writing”. Is this still accurate and up-to-date?

The formulation „at the time of writing“ is indeed unclear. I have addressed this concern by explaining the status quo of the policy. By April 2019 the policy  proposal was still in the policy process, the legislative process is assumed to be finished at the end of 2019. Following the ministry of agriculture, meat products with the label will be available in supermarkets by the year 2020.

Line 368. “will focus on initiatives of the retail sector”. I would have expected this much earlier in the paper, either in the introduction or in the methods section.

This has been changed accordingly.

Line 454 and beyond. The discussion on challenges is interesting, but is perhaps better placed in the discussion section. 

This has now also been included in the literature review.

While the paper overall is well written, it could benefit from editing by a native English speaker. 

To improve readability and expression the paper has been given to an English native speaker for correction.

Reviewer 2 Report

Extensive editing of English language and style is required - suggest proof-reader proficient in English language

Some English errors have been noted in first 4 pages (attached) - suggest serious review

"whereas" has been used too many times to start a sentence - suggest rearranging sentences

The manuscript is formatted as an article - suggest reformatting as a review

Clarification/definitions required for 'public' and 'private' actors (government versus private companies?) - particularly in abstract and simple summary

Definition/clarification of 'co-governance' needed

Define: "exceptional" agricultural policy

References missing for key points in introduction

Suggest inclusion of more subheadings

L107 - how are these two concepts different?

L198-201 - need to recognise limitations of Five Freedoms (cf. Five Domains). Five Freedoms are also not used as definitions of animal welfare

Author Response

Letter to Reviewer 2

Dear Reviewer,

Thank you very much for your detailed and very constructive review. Your suggestions have made a valuable contribution to the improvement of my study. In addition to the revisions in my manuscript, please allow me to respond to your comments. 

Kind Regards

Reviewer Comment

Author Response

Extensive editing of English language and style is required - suggest proof-reader proficient in English language

To improve readability and expression the paper has been given to an English native speaker for correction.

The manuscript is formatted as an article - suggest reformatting as a review

I have restructured the manuscript and I have tied it more closely to the theoretical debate on co-governance. In section 2.1. I introduce the literature on co-governance and then connect this debate to the literature on private policies in the field of farm animal welfare policies (section 2.2.). The hypotheses presented in section are then tested against two empirical cases (Germany and France). Given these changes in the structure and the content of the manuscript, I believe it now corresponds more to the usual structure of an article than to the structure of a review.

Clarification/definitions required for 'public' and 'private' actors (government versus private companies?) - particularly in abstract and simple summary

When I write public actors I refer to governmental actors. By private actors I mean primarily business actors such as retailers. The terms public and private are commonly used in the literature on co-governance, which is why I use them in the manuscript also. In the abstract and the introduction I now distinguish between governmental and market actors. 

Definition/clarification of 'co-governance' needed

I agree that the term co-governance has not been explained sufficiently in the original version of the manuscript. To address this concern I have added section 2.1. that presents the theoretical idea of co-governance, which is a relatively recent debate within the governance literature. Co-governance includes the dynamic interplay in the governing of common goods between public and private actors. Contrary to other governance forms such as corporate social responsibility, that is limited to business actors, co-governance always implies some sort of interplay between public and business or societal actors. Recent empirical work suggests that whereas co-governance is mostly cooperative it can likewise be competitive or conflictual and the relationships can change over time. In the application to the empirical cases (farm animal welfare policies in France an Germany) the analysis uncovers an increase in cooperative co-governance in addition to private governance. 

 Define: "exceptional" agricultural policy

In section 2.2. a short explanation of exceptionalism and post-exceptionalism has been added. Exceptional agricultural policies focus on the increase of productivity and the safeguarding of farmer’s incomes. Within this paradigm it is assumed that  governmental actors are key in securing these aims. This paradigm is recently being challenged by new ideas and interests. So-called post-exceptionalism includes new ideas emphasizing environmental sustainability, climate change, rural development or ethical considerations and as well issues concerning farm animal welfare.

References missing for key points in introduction

The introduction has been revised and in addition more references have been added.

Suggest inclusion of more subheadings

To improve the structure of the manuscript I have added subsections to the analysis (e.g. 2.1./2.2.) which I believe as well improves the reading flow.

L107 - how are these two concepts different?

I am not sure which concepts you refer to. Could you please give me a short note on what you mean so I will be able to clarify that? 

L198-201 - need to recognise limitations of Five Freedoms (cf. Five Domains). Five Freedoms are also not used as definitions of animal welfare.

The section on the Five Freedoms has been expanded and now includes a reference to the criticism that these freedoms concentrate solely on suffering and needs. In 2009 the British FAWC proposed that all farm animals should have a life worth living and an increasing number should have a good life. Indeed policymakers on the EU level refer to the original five freedoms in the development of farm animal welfare policies. However, even in countries with comparatively elaborated animal welfare regulations, the five five freedoms are often not guaranteed, let alone with the ambitious goals of a life worth living or even a good live. 

Round 2

Reviewer 2 Report

Thank you for your careful consideration of my earlier comments. This is a good review of the differences between regulations and welfare labels in Germany and France.

Some additional comments:

Please reformat this as a Review or Commentary. There is no original data or methods section for this to be an original article. See the Animals 'Instructions for authors'. I suggest a Commentary format for this manuscript.

I have made some small changes throughout the manuscript (please see attached) to assist you with reformatting this as a review/commentary article.

Please revise the Five Freedoms in line with their actual wording. "Freedom from.." not "Freedom of.." and a number of The Freedoms are written incorrectly e.g. (1) should be "thirst, hunger, or malnutrition"

Please capitalise 'Five Freedoms' 

I question the inclusion of the second row in Table 4 and the paragraph from Lines 486 to 497. This is not a labelling scheme and should not be recorded as such. Suggest removing and instead including a sentence at Line 511 to explain "Initiative Tierwohl"

Author Response

Dear Reviewer,

Thank you for the detailed reading of the manuscript and the very helpful suggestions.

Please allow me to respond to your suggestions below:

Regarding the five freedoms, you are right, that I did not define them precise enough. I have now included the original wording from the Farm Animal Welfare Council from 1965: (1) freedom from thirst, hunger or malnutrition; (2) appropriate comfort and shelter; (3) prevention, or rapid diagnosis and treatment, of injury and disease; (4) freedom to display most normal patterns of behaviour, and (5) freedom of fear. This wording has since then been adapted in numerous ways, an overview can be found in the report of the FAWC from 2009, which as well proposes the extension of the five freedoms. The European Union has proposed a different wording again, on their webpage they define the five freedoms as follows: Freedom from hunger and thirst,Freedom from discomfortFreedom from pain, injury and diseaseFreedom to express normal behaviourFreedom from fear and distress.

Regarding the table with the labels, I agree that the „Initiative Tierwohl“ should be deleted here, because it is not a label but a private assistance scheme. I have explained this in the text now.

Regarding the reformatting of the article into a review: I do not fully agree with that proposal, please allow me to explain my understanding of the issue: From my point of view the manuscript is more than a literature review because it does provide empirical data: the comparative analysis of public policies and of recent policy changes as well as the collection of the existing private animal welfare labels and the comparison with public policies. As to my knowledge, there are no policy analytical studies yet, that collect and investigate these very recent developments of public and private farm animal welfare policies and the interactions between these in the two countries. Therefore, within my discipline, this would clearly be an original article. I am fully aware that my understanding might deviate from the understanding in other disciplines. Given that animals is an interdisciplinary journal, I would of course be willing to do the suggested reformatting if the reviewer and the editors think that this would better fit the audience.